# Transcriptional Analysis of the Endostyle Reveals Pharyngeal Organ Functions in Ascidian

**DOI:** 10.3390/biology12020245

**Published:** 2023-02-03

**Authors:** An Jiang, Wei Zhang, Jiankai Wei, Penghui Liu, Bo Dong

**Affiliations:** 1Fang Zongxi Center, MoE Key Laboratory of Marine Genetics and Breeding, College of Marine Life Sciences, Ocean University of China, Qingdao 266003, China; 2Laoshan Laboratory, Qingdao 266237, China; 3Institute of Evolution & Marine Biodiversity, Ocean University of China, Qingdao 266003, China

**Keywords:** endostyle, RNA sequencing, *Styela clava*, ascidian, organ-specific gene, cross-species comparison

## Abstract

**Simple Summary:**

Formation of the complex organ has long been an arresting topic in evolution. The prototype of the complex organ is more likely to exist in creatures of ancient evolutionary hierarchy. In sister groups to vertebrates, including cephalochordate and urochordate, and the basal vertebrate, cyclostomate, a specialized pharyngeal organ is uniquely possessed by animals in these groups, named the endostyle. The endostyle serves as a filter-feeding and thyroid hormone-synthesizing function and is recognized as a prototype of the thyroid gland. Here, we comprehensively uncovered the expression profile of the endostyle in *Styela clava*, a wildly distributed ascidian with strong vitality. Cross-species comparison assures the function of the endostyle as a thyroid and parathyroid gland and the key composition of the digestive tract. Meanwhile, the high correlation with blood-producing organs, such as the head kidney and bone marrow in advanced vertebrates, implies a potential function of the endostyle.

**Abstract:**

The endostyle is a pharyngeal organ with an opening groove and cilia in invertebrate chordates (amphioxus and ascidian) and cyclostomate (lamprey), serving as a filter-feeding tract and thyroid-secreting location. Emerging evidence implies its complex cellular composition and potentially versatile functions. Multiple cell types in the endostyle have been thought to be progenitors of complex organs in advanced vertebrates. To describe the expression profile and the potential functions, bulk RNA sequencing on the endostyle in ascidian *Styela clava* was conducted and distinct markers were selected by multileveled comparative analysis. Transcriptional data assay and qRT-PCR-verified results showed the regional expression patterns of Hox genes in the longitudinal axis. Organ-specific markers of the endostyle was proposed by comparing expression with the main organs of the ascidian. A cross-species transcriptional profile projection between the endostyle and organs from *Danio rerio* and *Homo sapiens* indicates a robust homogenous relationship to the thyroid and digestive system of the endostyle. The high similarity between the endostyle and the head kidney in zebrafish/the bone marrow in human implies uniquely profound functions of the pharyngeal organ in proto-vertebrates. Our result revealed that the transcriptional profile of the human parathyroid gland was similar to the ascidian endostyle, indicating the evolutionary origin of vertebrate hormone secretion organs.

## 1. Introduction

The endostyle is a specialized pharyngeal organ existing in evolutionary transitional groups between invertebrates and vertebrates, including urochordates, cephalochordates, and ammocoete in Cyclostomata. The endostyle is a grooved organ partially encircled by ciliated epithelia tissue and longitudinally extended through the ventral wall of the pharynx [1]. In ascidian, the cross-section of the endostyle is a bilaterally symmetric structure composed of nine condensed epithelial compartments named by sequential number by relative position [2]. The groove facing the pharynx cavity functions as a water-filtering tract. Motile ciliates on the apical surface of epithelial tissue can motivate the water filtering and the mucus protein secreted from glandular tissue can primarily process food particles. The endostyle connects the oral region and esophagus morphologically as a part of the pharyngeal sac adjacent to the ventral blood vessel [3].

Besides the mucus secretion for food particle packaging and digestion, research has shown certain epithelial cell clusters are capable of concentrating iodine [4,5,6,7,8]. Expression patterns of key thyroid functional gene homologs, including thyroid peroxidase-like (TPO) and thyroid transcription factor-1 (TTF-1), are mainly distributed in the dorsolateral part of the endostyle, proving the thyroid-resembled function of the endostyle [9,10]. More interestingly, with the morphogenesis of ammocoete, young endostyle will transit into thyroid follicle cells, which strengthens the evidence of the endostyle as a thyroid gland archetype [11]. Collectively, for nine regions of adult ascidian endostyle cross-section, zones 1, 3, and 5 are supporting regions, zones 2, 4, and 6 are glandular regions, and zones 7 and 8 are regarded as thyroid homolog tissue. Meanwhile the amphioxus contains seven compartments, including supporting zones 1 and 3, glandular region zones 2 and 6, and iodine-concentrating region zones 5a, 5b, and 6 [9].

During the process of individual development, the endostyle is generally believed to be originated from the endoderm [12]. Different transcriptional factors are expressed in distinct zones. TPO, TTF-1, Pax2/5/8, FoxE4, FoxQ1, and FoxA have distinct regionalized expression patterns of tissue compartments [9]. Meanwhile, orthodenticle homeobox (Otx), Pax2/5/8, and Hox1 show regionalization of anterior-posterior expression patterns [13]. However, the whole blueprint of the architectural formation of the endostyle remains incomplete. Recent studies on tunicates showed that the perturbance of two key transcriptional factors in endostyle development, NK2 Homeobox 1 (Nkx2-1) and FoxE, leads to different effects on the developing endostyle, implying the divergence of a glandular (mucus-secreting) and thyroid-tissue element regulatory network [14,15].

As the particularity and novelty of the endostyle are possessed only by non-vertebrate chordates and ammocoete in Cyclostomata, the evolutionary status of the endostyle has long been discussed along with the origin of the thyroid organ. In lamprey larvae, the transformation to thyroid follicles during endostyle metamorphosis gives a recapitulation of thyroid organ emergence in vertebrate evolution, which makes it acknowledged that the evolutionary origin of the thyroid gland lies in endostyle epithelial compartments [16]. However, a detailed investigation of the hagfish, another existing cyclostome other than the lamprey, discovered the absence of an endostyle structure and the thyroid was directly developed from the pharyngeal floor [17]. A new evolutionary scenario was proposed then that the hagfish share a common ancestor with gnathostomes, while the endostyle is a lamprey re-acquisition [17]. Furthermore, whether the endostyle is a plesiomorphic trait of vertebrates needs to be confirmed with more evidence.

Immune activity is also a key function in the endostyle as the direct exposure to seawater. Kinds of immune effector cells with certain markers were defined in the endostyle and subendostylar region bathed with circulatory cells. In solitary ascidian, Toll-like receptor 2 and vasoactive intestinal peptide and lectin histochemistry were used in defining immune zone cells and proved zone 8 with a multilabeled immune function [18]. Additionally, galectins were proved to be secreted by the endostyle under the challenge of lipopolysaccharide (LPS) [19].

The colonial ascidians have a strong capacity for asexual reproduction and stem-cell mediated development process. Research on ascidian *Botryllus schlosseri* has proved the anterior ventral region of the endostyle and adhering blood cells from the subendostylar sinus as an adult somatic stem cell niche that supports the organ regeneration and development of new zooids [20]. Along with the adult endostyle of the colonial ascidian, a niche of somatic stem cells was discovered and termed a cell island [21]. The endostyle niche in *B. schlosseri* was believed to be homologous to the hematopoietic bone marrow of advanced species as the discovery of hematopoietic stem cells, blood cells, and immune effector cells in the endostyle niche [22]. The research above gives a detailed description of the evolutionary status of the subendostylar sinus region. As the closest relationship of tunicate to vertebrates [23], the research on the endostyle of ascidian indeed gives a down-to-earth relationship in evolution.

By comparing multi-organ within-species and cross-species organ transcriptional profiles, the endostyle was defined as having a high similarity to other vertebrate tissue [24]. Previously reported endostyle-specific genes, thyroid-related transcriptional factor genes, and immune-response genes were defined as organ-specific genes (OSGs), consistent with the previous report. Intriguingly, the neural-specific developmental gene *Slit* was specifically expressed in the endostyle [24]. All the new insights on the endostyle functions, including immune functions, stem cell activity, and neural marker expression, imply that the endostyle is likely to be an organ with versatile functions.

Here, we utilized the bulk RNA sequencing data on the endostyle in ascidian *S. clava*, whose genome has been sequenced and whose transcription factors have been deeply investigated [25,26]. The morphology of endostyle segments was described and compared with the cross-section. The basic function of the endostyle was comprehensively described with gene ontology (GO) analysis and top functional gene selection. Genes with functions of thyroid hormone synthesis, immune response, digestion, and neuron process were defined. Cross-species comparison with *Ciona robusta* was conducted in defining the OSGs in the endostyle by comparing the expression profile of the endostyle with other organs. Based on the OSG, similarities between the endostyle and the organs of *D. rerio* and *H. sapiens* were calculated and interpreted.

## 2. Materials and Methods

### 2.1. Animals and Sample Preparation

The adult ascidian *S. clava* was collected from the coastal areas of Weihai City, Shandong province (122.41° E, 37.16° N). The ascidian was temporarily persevered in the aquarium system in the laboratory at a temperature of 18 °C with a consistent oxygen supply. The healthy animal was dissected manually with disinfected instruments initiated by the removal of the stem tunic part and scissored from the ventral body mid-line. The endostyle was gently dissected from the dorsal pharyngeal wall and segmented into three longitudinal parts of the same length for the convenience of the following experiments. Three individual ascidians were sacrificed for the endostyle collection. The guidelines for animal experiments were approved by the Ocean University of China Institutional Animal Care and Use Committee (OUC-IACUC) with approval number 2021-0032-0027.

### 2.2. Preparation of Cryosection and HE Staining

To visualize morphological features of the endostyle cross-section, a cryosection preparation and staining technique was carried out on the endostyle sample. Fresh endostyle tissue was carefully dissected from the ascidian and immediately immersed into the precooled 4% paraformaldehyde (PFA) and persevered on a gentle shaker table for 16 h at 4 °C. The tissue after fixation was washed with cooled PBS to wash off PFA and the tissue was transferred into 30% sucrose solution (made with PBS solution) in the flat-bottomed container. All processes of fixation and sucrose sinkage were conducted at the temperature of 4 °C. After full sinkage, the solution on the surface of the tissue was dried with Kimtech wiper (FIS: 06666) and the tissue was merged into the optimal cutting temperature compound (OCT, SAKURA Tissue-Tek^®^ O.C.T. Compound) in the model of a proper shape. Liquid nitrogen was used to snap-freeze the block. The embedded block after freezing was fully covered by aluminum foil and preserved in sealed containers at −80 °C.

Cryosection was prepared with Leica CM3050 S Research Cryostat. Temperature pre-balance was performed for 30 min before sectioning the frozen block. The cryosection was pasted onto adhesion microscope slides and persevered at −80 °C. The HE staining was accomplished following the procedure of the Modified Hematoxylin and Eosin (HE) Stain Kit from Solarbio (G1121).

### 2.3. RNA Extraction and Transcriptome Sequencing

The total RNA of the endostyle samples was extracted with RNAiso Plus (Takara, Code No. 9108). The integrity and quality of the total RNA was evaluated by agarose gel electrophoresis and Nanodrop spectrophotometry (Eppendorf). Before library construction, the quality of RNA was finally analyzed by Agilent 2100 bioanalysis. The library construction and sequencing experiment were accomplished by the Novogene experimental department. mRNA was enriched and the library was constructed with NEBNext^®^ Ultra™ RNA Library Prep Kit for Illumina^®^. The library was sequenced on an Illumina Novaseq platform and 150 bp paired-end reads were generated. Clean reads were obtained from Novogene.

### 2.4. Transcriptome Analysis

Clean reads were aligned against the reference genome with hisat2 [27]. Results were produced in the SAM format. Samtools [28] were used in bam file transferring and bam file sorting. Based on the sorted bam file, String Tie [29] was used in read assembly and quantitation of transcripts with the ballgown flag activated. The transcriptional count file was produced with official prepDE.py2 script (http://ccb.jhu.edu/software/stringtie/index.shtml?t=manual#deseq, accessed on 31 May 2022). The FPKM and TPM results were transferred based on the count file. Differential expressing genes were analyzed with R package DEseq2 [30]. All three segments’ data were compared in pairs. Genes with a p-adjust value lower than 0.05 and log2FoldChange bigger than 1 or lower than −1 were defined as differential expressed genes (DEG). DEG was visualized on the Volcano plot and heatmap plotted by ggplot2 [31] and pheatmap (https://CRAN.R-project.org/package=pheatmap, accessed on 31 May 2022). Gene enrichment analysis was performed with clusterProfiler [32]. Top functional genes were manually selected from the top 300 expression genes in the endostyle.

### 2.5. Quantitative Reverse-Transcription PCR (qRT-PCR)

qRT-PCR was used in verifying the expression of DEGs and the transcriptome results. The cDNA of the endostyle was synthesized by reverse transcription reagent, HiScript II Q RT SuperMix for qPCR (Vazyme, R223-01). The primers for qRT-PCR were designed utilizing the NCBI primer design tool (https://www.ncbi.nlm.nih.gov/tools/primer-blast/, accessed on 2 November 2022) (Table 1). The uniqueness of all target sequences was verified against the *S. clava* genome. RT-qPCR was performed using the SYBR Green PCR Master Mix (Vazyme, Q411-03) on Light Cycler 480 (Roche). All RT-qPCR primers were listed in Table 1. EF1α was used as the reference gene. Data were calculated using the 2^−ΔΔCt^ method. Visualization of qRT-PCR results was carried out with R package ggplot2 and ggsignif (https://rdocumentation.org/packages/ggsignif/versions/0.6.4, accessed on 11 November 2022).

### 2.6. OSG Identification

Transcriptome data of *C. robusta* organs were obtained from a published dataset [24], from which a gene expression matrix of 67,716 transcriptional features were used for the next step of analysis. The public ID of the published dataset was PRJNA731286 in the NCBI bio-project. Cross-species homolog gene alignment was conducted with localized BLASTP, in which the longest transcript for each gene in *Styela* was aligned against the transcript dataset of *C. robusta* and the max_target_seqs was set to 1. The expression matrix of *C. robusta* in reads per kilobase per million mapped reads (RPKM) format was merged with *Styela* endostyle expression profile in fragments per kilobase per million mapped reads (FPKM) version. Homolog gene pairs with an expression value larger than 0.5 in *Ciona* organs (except endostyle) were eliminated. Genes with expression values larger than 1 in *Styela* endostyle were defined as OSGs of *S. clava* endostyle. A total of 104 OSGs were obtained (Appendix A).

### 2.7. Endostyle OSG Projection on Zebrafish Organ Transcriptome Profile

OSGs in the endostyle were aligned against the zebrafish (*D. rerio*) genome (Version GRCz10) with BLASTP. One hundred unique pairs of homolog genes between zebrafish and *Styela* endostyle OSGs were obtained. To obtain as many organs expressional data as possible, multiple transcriptomic datasets were involved in the analysis. The raw FASTQ files of organs such as tail, spleen, skin, muscle, liver, heart, and brain were obtained from GSE171906 [33]. The raw FASTQ files of organs, including the head kidney and gill, were obtained from SRP044781 [34]. The expressed counts and TPM matrix of organs above were processed in methods in 2.4. For the gene expression value (TPM) of the thymus organ of zebrafish, raw data in FASTQ format were obtained from GSM5732077 [35]. The raw data were aligned against the genomic with Cell Ranger. The resulting filtered matrix was obtained and applied with the sum function by row in obtaining the count of all cells per gene. The count table was transferred to the TPM value with the length of every effective transcript (exon length). The thyroid data of the zebrafish were obtained from GSE133466 [36]. TPM values were obtained in the same method for the thymus data. For 100 homolog gene pairs for endostyle OSGs in *D. rerio*, genes with TPM larger than 1 in *D. rerio* organs were defined as identical expressed genes. The ratio of highly expressed genes in each organ of zebrafish was defined as a similarity ratio.

### 2.8. Transcriptional Profile Comparison between the Endostyle and the Potentially Similar Organs

The transcriptional matrix of the endostyle and pharynx, gill, head kidney, and thymus were compared manually with the Venny online interactive tool (https://bioinfogp.cnb.csic.es/tools/venny/index.html, accessed on 25 January 2023). Shared and organ-only expressed gene lists were utilized for GO enrichment analysis with the R package clusterProfiler [32].

### 2.9. Similarity of the Endostyle Transcriptome Profile to Human Organs/Tissues Enriched Gene Set

Human tissue-enriched genes were obtained from the protein atlas (https://www.proteinatlas.org/about/download, accessed on 25 January 2023). RNA consensus tissue gene data were obtained from the website, which included the consensus transcript expression levels summarized per gene in more than 50 tissues of the human. Twenty-seven main tissues were taken into account. The tissue-enriched gene was defined with an expression level of at least a four-fold higher mRNA level in a particular tissue compared with any other tissue (https://www.proteinatlas.org/humanproteome/tissue/tissue+specific, accessed on 24 January 2023). The protein sequence of all tissue-enriched genes (in UNIPROT gene id) was aligned against all proteins of *Styela*. Based on the homolog correlation, all paired homolog tissue-enriched genes were evaluated with the expression level in the endostyle. Homolog pairs with FPKM larger than 5 were defined as positive-expressed human-tissue-enriched genes in the endostyle. The similarity index was the ratio of the positively expressed genes in each list of tissue-enriched genes for human organs. The human anatomical graph was plotted with gganatogram [37].

## 3. Results

### 3.1. Morphological Observation and Expression Profile of the Endostyle

The endostyle in the ascidian is longitudinally extended on the dorsal pharyngeal wall (Figure 1A). The cross-section of the endostyle is a groove-like channel with water flooding through. Cross-sections based on the cryosection technic were stained with HE staining (Figure 1B). The basic structure of the cross-section was similar among longitudinal levels, including compartments of the epithelial region bathed in the surrounding extra-cellular matrix. Long cilia born in zone 1 were seen, which can collaborate with ciliates in other zones constructing the mucus net and facilitating the filter-feeding process [38].

To comprehensively uncover the function of the endostyle, bulk RNA sequencing on the endostyle was conducted. For the convenience of sample collection, each endostyle was evenly divided into three continuous segments. Three biological replicons were collected for RNA extraction and sequencing. All samples were of similar expression distribution (Appendix A). High similarity within individual endostyles is more significant than inter-individual correlation, i.e., three segments from endostyle no. 1 had a correlation index of 0.98 between A, B, and C segments. Principal component analysis on nine endostyle samples showed three main groups corresponding to three endostyle individuals (Appendix A). The similarity identities of these samples showed clear individual similarity rather than segment discrepancy, which gives fundamental information of uniformity within the endostyle longitudinal level.

Based on the expressed profile of the endostyle, GO enrichment analysis on expressed genes (FPKM larger than one) was conducted (Figure 1C). Categories according to filter-feeding and the secreting process can be seen, including protein processing-related pathways (protein phosphorylation and translation), and secreting-related processes (vesicle-mediated transport). Some cellular component processes were also consistent with structure features, including constituent of the ribosome and dynein complexes.

Meanwhile, the top 300 expressed genes of the endostyle were filtered and selected manually based on the functional annotation of the genes. The functions of four main categories were selected and the relative expression level was visualized by clustered heatmap (Figure 1D). For example, the function of thyroid hormone synthesis of the endostyle has long been believed to be the evolutionary role of the endostyle [16]. Relating genes were also seen in the top expressed gene list, including the TPO and dual oxidase maturation factor 1-like (DUOX1) gene, the high expression of which is consistent with reports of thyroid hormone synthesis capacity [39]. Potential related organic transporters, such as the sodium-coupled monocarboxylate transporter, were also detected in the transcriptional profile, while the detailed role remains controversial [40].

The immune function of the ascidian exclusively relies on innate immunity [41]. As a pharyngeal organ directly exposed to seawater, the capacity of immunity was also detected in the endostyle. Broad-spectrum anti-microbe peptides clavanins A and D were intensively detected in the endostyle. Complement system factor C3 and intelectin 1 and 2, which are glycan-binding proteins [42], were also seen in the marker list. As a transcriptional factor downstream of Toll-like receptor, an interferon regulatory factor was also detected in the transcriptional data (Figure 1D).

As an organ-facilitated filter-feeding [43], the endostyle also has a digestive functional gene expressed. Broadly-distributed mucus functional protein mucin 5AC was detected. The expression of neurosecretory functional genes [44] also was seen in the endostyle expression profile, including neuroserpin, semaphorin 1A, neuronal cell adhesion molecule (NCAM), contactin-associated protein 4, and reelin (Figure 1D).

### 3.2. Longitudinal Distinct Genes of the Endostyle

Differential expression analysis was conducted on the longitudinal levels of the endostyle. The expression profile of each two segments was compared to obtain DEGs. The volcano plot shows DEGs of each two comparing groups (Appendix A). The filtering scale of log2FoldChange was set to one and the p-adjust value was set to 0.05. To analyze the expression trend on the longitudinal level of DEGs in detail, all DEGs selected by differential gene expression analysis were listed with a heatmap (Figure 2A).

A few genes with expressional longitudinal change gradients can be seen in the heatmap (Figure 2A). For example, the protein rolling stone, which is a key factor in muscle development and function maintenance [45], shows a decrease in expressing gradient along the anterior to the posterior body axis. The trend of the protein rolling stone reflects a relatively strong function of muscle in the anterior part of the endostyle. A few genes with distinct expression in the anterior part of the endostyle were also detected in DEG analysis, including hemicentin 1, sushi, von Willebrand factor type A, EGF and pentraxin domain-containing protein 1 (SVEP1), P selectin, epiplakin, matrillin 2, high expression of which may indicate the intense capacity for the epithelial/endothelia-ECM interaction process in the anterior part of the endostyle [46,47,48,49].

Meanwhile, some genes with expression patterns enriched in the posterior part of the endostyle were also defined (Figure 2B). Here, we detected a significant divergence of Hox gene expression in endostyle segments consistent with previous research in *C. robusta* juvenile [50], which is highly expressed at the posterior region of the endostyle. Both Hox A3 and Hox A4 had the highest expression at the posterior part of the endostyle, while the Hox A3 had a higher expression level in the middle segment of the endostyle. The qRT-PCR result also showed for Hox A3, the anterior part had a significant difference compared with the middle part, while for Hox A4, a significant difference was not seen.

### 3.3. OSG-Based Expression Similarity Analysis

OSGs are important in understanding organ development and function. Under the circumstances of a lack of organ transcriptomic data from *Styela*, to decipher the function of endostyle in the ascidian we choose to use Bulk RNA sequencing data of 11 organs or tissues from adult *C. robusta* as a reference dataset to identify OSGs of the endostyle [24]. The organ or tissue from *C. robusta* includes the oral siphon, atrial siphon, neural complex, heart, ovary, pharynx, stomach, proximal intestine, middle intestine, distal intestine, and endostyle, which are the main organs of an ascidian adult individual. Unique transcripts with the longest length of *S. clava* were aligned against the total protein sequence of *C. robusta* with BLASTP for obtaining comparable homologous gene pairs. A total of 18,203 homolog transcript pairs were obtained as the cornerstone of two ascidian cross-species’ comparison. All gene or transcript functional annotations were adopted from the version of *C. robusta* (Figure 3A). In the comparison step, all gene pairs that have expression values larger than 0.5 were ruled out in organs or tissue of *C. robusta*, except the endostyle. Based on the gene pair set above, for the endostyle of *S. clava*, the gene with an expression level higher than one was defined as an endostyle-specific gene (Appendix A). A total of 104 OSGs with clearly functional annotations of the endostyle were identified compared with 11 *Ciona* organs or tissues.

As the selected result (genes with distinct functional annotation) showed (Figure 3A), some genes have parallel expression levels in the endostyle of both *C. robusta* and *S. clava*. For example, cationic amino acid transporter 2 (CAT2) has a significantly high expression in the endostyle of both species. The expression of B-cell CLL/lymphoma 6 member B protein (BCL6B) was also detected in both species, which was reported as a functional factor related to early B-cell development events in the mammal model [51]. The high expression of fibropellin was also seen in both species’ endostyle, which caters to the structure feature of the endostyle. The neuron-specific transcriptional factor, *slit* homolog, was also detected.

To probe the general correlation between the endostyle and the organs in advanced species, a comprehensive cross-species expression profile similarity analysis between the endostyle and organs from *D. rerio* was conducted (Figure 3B). For all 104 OSGs of the endostyle, alignment against the zebrafish genome was conducted and 100 homolog gene pairs were obtained. The expression profile of all 100 homolog genes in all 12 zebrafish was summarized (Appendix A). Genes with expression values (TPM) larger than one were defined as identically-expressed genes in an organ. The similarity ratio equals the ratio of identically expressed genes in all homolog genes. As the result showed, the gill, head kidney, and brain of the zebrafish have the highest expression similarity ratio of homolog OSGs. The gill in the bony fish served as a water-filtering and oxygen-intake functional organ, which is also an essential pharyngeal organ. The brain is the central neuron organ, while the head kidney is analogous to the mammalian bone marrow and the primary site of definitive hematopoiesis [52]. The data provide a general recognition that the endostyle shares a strong similarity to neuron and hematopoietic systems.

### 3.4. Transcriptomic Profile Comparison Reveals Features of the Endostyle

The expression of OSGs in organs and tissues of advanced species may reflect the evolutionary lineage of homolog organ functions in animals [24]. The potential high transcriptional similarity between the endostyle and the organs in the *D. rerio* was exhibited based on the OSGs of the endostyle. However, to what extent and functional categories the endostyle is similar to potential homolog organs still needs to be revealed.

As the anatomy adjacent location between the endostyle and the pharynx sac possessing gill slit, a transcriptional-profile-based organ functional comparison was conducted between the endostyle of *Styela* and the pharynx of *Ciona*. Bilateral BLASTP was conducted in constructing homolog correspondence between the two species. For expressed genes of two species (genes with FPKM value larger than one), 5528 shared-expressed homolog gene pairs were detected, while 1254 and 1725 genes were uniquely expressed by either the pharynx or the endostyle (Appendix A). To reveal potential functional similarities and differences between these two organs, GO enrichment analysis was conducted on three expressional categories (Appendix A). As the enrichment results showed, the endostyle and the pharynx have shared enriched terms for some housekeeping cellular components, including microtubule-based movement, extracellular matrix organization, and so on. However, for organ-specific GO terms, the endostyle gene group hits on terms including ion transport and ion channel activity, suggesting potential distinct osmotic pressure balance or neural signal transport cellular activity in the endostyle. The pharynx enriched the term, dopamine beta-monooxygenase activity, suggesting the distinct capacity of neurotransmitter-synthesizing compared with the endostyle.

In Section 3.3, the OSG-based expression similarity ratio was calculated in describing the potential similarity between the endostyle and the organs from the advanced aquarium species, *D. rerio*. Organs with a top-ranked expression similarity were the gill, head kidney, and brain (Figure 3B). Although the endostyle has long been believed to be a homolog organ to the thyroid [9], a few organs also showed a higher general expression similarity. To better characterize the potential similarity and difference between the endostyle and the organs with a high similarity ratio, a transcriptional profile-based organ functional comparison was conducted. The expression profile of the organ with a similarity ratio of up to 70 percent, the gill, was compared with the endostyle. A total of 5119 shared expressed genes were detected. A total of 539 and 735 organ-specific expressed genes were defined for the *Danio* gill and the *Styela* endostyle, respectively (Figure 4A). GO analysis showed that shared functional terms between these two organs’ transcriptional profiles were diverse, including central nervous system morphogenesis, intraciliary transport particle, digestive system development, gland development, cilium organization, and so on (Figure 4B). As two important pharynx organs, the shared terms of the gill and the endostyle showed a similar transcriptional profile of their potential functions.

In the comparison group between the *Danio* head kidney and the *Styela* endostyle, a similar comparison strategy was conducted (Figure 4C). Shared GO enrichment analysis terms include cilium or flagellum-dependent cell motility, myeloid cell differentiation, erythrocyte homeostasis, and so on, suggesting similar expressional functions of these two organs (Figure 4D). Meanwhile, many endostyle-only terms were seen, including gland development (Figure 4D) and endocrine system development, paralleled with the recognition of the gland and secreting function of the endostyle. A few terms were seen in endostyle-only terms of both gill–endostyle and head kidney–endostyle comparison groups, including inner ear receptor cell development and sensory perception of the mechanical stimulus, suggesting a potential unique function of the endostyle on these terms.

The endostyle was also believed to possess an immune function in previous reports [19,22,53]. Here, we utilized the comparative analysis strategy and compared the expression profile of the *Danio* immune organ, the thymus, and the *Styela* endostyle (Appendix A). In enrichment results of the shared expression genes, terms including maintenance of gastrointestinal epithelium, erythrocyte homeostasis, vacuolar transport, and myeloid cell differentiation were enriched. However, terms relating to immune function were not detected, which may attribute to the different immune functional pathways between the thymus and the endostyle.

### 3.5. The Expression Similarity of Human OSGs Homolog in the Endostyle

As a pharyngeal organ with generally believed filter-feeding and thyroid hormone synthesis capacity, multiple potential functions that are homolog to other organs of advanced animal have long been a target of potential functional research of the endostyle. Here, more than 3000 tissue-enriched genes of 27 organs were obtained from the protein atlas of *H. sapiens* [54]. More than 3000 tissue-enriched genes were aligned against the protein file of *S. clava* and around 2700 ortholog gene pairs were obtained. For the tissue-enriched gene set of a certain organ, if the expression level of their homolog gene in *Styela* was larger than five, the gene was defined as a similarly expressed gene (Appendix A). The ratio of similarly expressed genes in the tissue-enriched gene set of the organ was calculated and defined as the similarity index between the human tissue and the endostyle (Figure 5A).

As the result showed (Figure 5A–C), the thyroid and the stomach are two organs with the highest similarity index of the endostyle, which is consistent with previous recognition of the thyroid hormone synthesis and the pharynx digestive track. Parathyroid was also a highly ranked term in the similarity index, showing the transcriptional similarity of parathyroid with the endostyle, which has not been reported before. The salivary is the number four ranked organ, which is parallel to the character of the mucus-secreting function of the endostyle. In the blood system of the eyes in humans, the choroid ranked number five. The subsequent fallopian and skin systems are both mainly composed of the epithelium, similar to the expression of the condensed epithelium tissue of the endostyle [1]. To summarize, other than the expression similarity to the thyroid organs and the digestive and secreting organs, including the stomach and salivary systems, the parathyroid is for the first time reported to share a high expression similarity to the endostyle. The expression similarity to choroid and bone marrow also hints at the potential role of the blood system and hematopoiesis of the endostyle.

## 4. Discussion

### 4.1. Morphological and Functional Definition of the Endostyle

As a pharyngeal organ uniquely possessed by protochordates and basal vertebrates, the endostyle has long been characterized as a mucus-secretory organ facilitating the filter-feeding process. Because of the regional thyroid hormone synthesis capacity and transformation into follicle thyroid-like cells during metamorphosis in ammocoetes, the endostyle is believed to be an evolutionary precursor of the thyroid gland [55]. However, recent evidence has shown the capacity of the innate immune process [18,19], stem cell [20], and hematopoiesis [22] in both the traditional dense regions and the subendostylar region of the endostyle. The traditional region of the endostyle is mainly focusing on the function of mucus-secretory capacity by epithelial tissue, while the diverse cellular composition and potential function of the subendostylar region are largely ignored. However, it is seen in the cross-section of the endostyle (Figure 1B) that the traditionally defined epithelium region 1–9 has not come to an end at the boundary of so-called zone 9. The epithelium has consistently extended until the dorsal wall of the region, which connects the tissue of the pharyngeal branch sac extending toward two lateral directions. Within the relatively enclosed region surrounded by single-layer epithelial, a large number of cell individuals are immersed in the highly developed extra-cellular matrix. It is natural to facilitate us to update the traditional definition of the endostyle with reported evidence, instead of the traditional eight-zone structure composed mainly of epithelium tissue [1], and both dense epithelium tissue and the surrounding sinus-like region are supposed to be defined as a pan-endostylar region. At the dorsal region of the pan-endostylar region, a vessel-like region could be seen, which is the so-called subendostylar vessel.

In this research, high-quality RNA-seq results were obtained with three biological replicons. The GO enrichment analysis on all expressed genes was obtained, implying the function of protein synthesis and the secretion process. A distinct cellular component, the dynein complex [56], was also detected as a highly enriched term. Based on prior knowledge reported before, genes in four functional categories, including the thyroid hormone synthesis function [9], the immune function [18], the digestive function [57], and the neuron function [44], were selected from the top 300 expressed genes.

For all four categories, the highly expressed genes can describe the fundamental functional pathway in the endostyle. Most top expressed genes in the immune function are of the innate immune pathway, which is consistent with previous research on the ascidian immune response [58]. Interestingly, the homolog factors of two adaptive immune response-related factors in vertebrates were also detected, which might become involved in alternative immune functional pathways in the endostyle immune response. For example, Src kinase-associated phosphoprotein may couple with the macrophage adhesion process and Caspase recruitment domain-containing protein 11 may become involved in signal transducing. Meanwhile, for the neuron functional categories, some of the markers were not reported before and need further verification. For example, neuroserpin [59], semaphorin, NCAM [60], contactin-associated proteins such as 4 [61], and reelin [62] are all key transcriptional factors or proteins in neuron development, maintenance, growth, and synapse construction.

### 4.2. Longitudinal Heterogeneity within the Endostyle

In this research, for the convenience of sample collection, three longitudinal segments were collected separately in ensuring the quality of total RNA extraction. Under this sample-collecting strategy, the general expression profile of the endostyle samples was similar (Appendix A), while longitudinal expression divergence of a few genes was observed with differential expression analysis (Figure 2A). Parallel to the previous report, the Hox A3 and Hox A4 had enriched expression patterns in the posterior region of the endostyle [50], which is a conservative mechanism in body architecture formation. Concretely, for the Hox A3 expression pattern, the qRT-PCR result (Figure 2B) showed that the expression level of the middle segments is significantly higher than the anterior segments, while no significant differences were seen in Hox A4, also notable in the RNA-seq heatmap (Figure 2A). The detection of the delicate difference was not only consistent with the previous report, which depicted that the expression region of Hox A3 had more anterior boundary than Hox A4 [50], but also proved the sensitivity of the RNA-seq data.

Interestingly, a few genes with distinct expressed trends for segments were also detected, including proto-oncogene c-Fos-like in the posterior part, and the protein rolling stone in the anterior part. GO information and the potential function of divergence expression patterns of these genes need further study and verification. For DEG filtered in the middle part of the endostyle, only two genes were concluded, which may reflect that most genes have longitudinal even expression trends or transiting expression patterns across longitudinal levels under the current sample collecting strategy.

### 4.3. OSG of the Endostyle Showed Distinct Expression Profiles Compared with Non-Endostyle Adult Organs in Ascidian

By comparing with the expression profile of *Ciona* organs, endostyle OSGs were selected, which portrayed the expression of distinct markers (Figure 3A). For example, CAT2 had a significantly high expression in the endostyle. It was reported that CAT2 is a very important regulator of the immune response during the *Helicobacter pylori* infection process in mammals [63]. The distinct expression of CAT2 in the endostyle strongly supports the pathogen-induced immune function of the endostyle as a part of the ascidian digestive system. The expression of BCL6B was also detected in the endostyle of both species (*Styela* and *Ciona*), which was reported as a functional factor related to early B-cell development events in mammal models [51].

Fibropellin, a family of extracellular matrix proteins, has the function of assisting epidermal growth and the differentiation of mesenchymal and epithelial cells. The high expression of fibropellin caters to the structure feature of the endostyle. The transcription factor homolog of aristaless, which plays a vital role in both the neuron system and pancreas development [64], is also specially expressed in the endostyle of both species. More interestingly, the slit homolog had a clear expression on both species. The slit gene is found in most bilaterians and is of key importance in the development of the nervous system [65]. OSGs defined in this analysis imply the distinct expression of endostyle in immune-functional genes, extracellular matrix components, and neuron system transcriptional factors.

### 4.4. Cross-Species Comparison Based on the OSGs

The formation of complex organs in advanced vertebrates has long been an attractive topic in evolution. For example, the evolution of the eye was thought to be “absurd to the highest degree” by Charles Darwin in *The Origin of Species*. Nowadays, we assure that complex eyes could be evolved from very simple ones with the driving force of natural selection. From this example, the origin or evolutionary node pattern of the complex organ is necessary for drawing a realistic evolving path in forming a complex organ in advanced vertebrates. The endostyle is only preserved by cephalochordates, urochordate, and larva lamprey (which is thought to be a homolog organ to the thyroid gland); more specifically, only zone 7 is capable of concentrating iodine. However, as a specialized pharynx organ shares the same body location as the pharynx of advanced animals, the endostyle is very likely to preserve another prototype of pharynx organs.

Here, by utilizing the transcriptomic comparison method, OSGs of the endostyle were obtained in *S. clava*. Based on these OSGs, a cross-species transcriptional profile comparison was conducted in probing the possible correlation of the endostyle to more advanced organs. From the perspective of the shared expression of the endostyle OSG homolog, the expressional similarity to organs in zebrafish was obtained (Figure 3B). The result showed a high similarity to the gill, head kidney, and brain, which is a general guidance in understanding the potential functions of the endostyle.

A highly developed gill is widely possessed by aquatic animals. For the ascidian in this research, the organ or tissue with the homolog function to the gill is the pharyngeal sac, which possesses rows of gill slits dominating the major volume serving the water-filtering process. Meanwhile the endostyle extends longitudinally through the ventral wall of the pharyngeal sac, collaboratively constituting the pharynx of the ascidian. Therefore, the general expressional similarity between these two anatomical systems is logical. We first utilized two *Ciona* organs’ bulk RNA-seq data (PRJNA731286) [24], the pharynx and the endostyle, to compare their shared and distinct expression. Summarizing the shared enrichment results (Appendix A), the pharynx and the endostyle have a shared immune function in maintaining homeostasis, the neuron system (including axon and voltage-gated channel in tissue), and heme transporter capacity involving the blood system function. Moreover, some shared cellular components, including cilium movement in cell motility and dynein/axoneme complex, showed a shared function of cilium movement for water-filtering (oxygen exchange and filter-feeding). Concluding the organ-distinct terms (Appendix A), the endostyle has a distinct function of synaptic connection and a sensory perception of sound, while the pharynx has a distinct function in the immune system (Toll signaling pathway and interleukin production). We also used the *Ciona* pharynx and the *Styela* endostyle to conduct a similar comparison and the enrichment functional terms are basically consistent with results of the pharynx–endostyle of *Ciona* (Appendix A), while relating terms are less because of the correspondence of different species in comparison.

In an OSG-based similarity ratio analysis, the head kidney is the second highest in organ similarity ranking. The head kidney in fish is believed to be analogous to the mammalian bone marrow and the primary site of hematopoiesis. The high similarity between the endostyle and the head kidney hints at potential functional similarity. In a transcriptional expression profile comparison analysis (Figure 4C,D), shared GO terms include myeloid cell differential and erythrocyte homeostasis, supporting the similar function index between the endostyle and the head kidney.

Meanwhile, the level of shared expression of the organ/tissue-enriched gene homolog of human was used as a similarity between human tissues and the endostyle. Interestingly, in this result, except for the generally acknowledged thyroid and digestive-related tissue (stomach and salivary), the parathyroid was reported for the first time as a highly similar organ, which needs further research and evidence in supporting the possible evolutionary lineage. Similar to the result of zebrafish, the blood system, and hematopoiesis corresponding organs, the choroid and bone marrow both have a high similarity to the endostyle. The choroid is a middle layer of tissue of the wall of the eye, which is composed almost entirely of blood vessels functioning as the blood supplier of the retina. The high similarity of the choroid and the endostyle showed potential similarity between these two organs, which is consistent with the research on the colonial ascidian endostyle [22]. These results double verified the possible similarity of the hematopoiesis capacity of the endostyle, which is in accordance with the research in the colonial ascidian [22]. The similarity to the retina implies the potential similarity of neuron-circuitry-involved signal conversion.

Cell types are the basic building blocks for multicellular organisms, which are also fundamental evolution units such as genes and species [66,67]. The development of cellular-scaled transcriptomics facilitated the identification of cell types in high throughput, which also bolstered the research of cell-type evolution. In this research, we proved a transcriptome correlation or similarity based on specific genes filtered from a comprehensive and high-qualified bulk RNA sequencing with relatively limited resolution. In future prospective research, understanding the cell lineage evolution from cell-type resolution is necessary.

## 5. Conclusions

Bulk RNA sequencing on the endostyle in *S. clava* provided a detailed profile of the endostyle function and the multileveled comparison selected certain markers in representing its distinct features. Basic functions of the endostyle, including thyroid hormone synthesis, immune function, digestion, and neuron function, were supported by distinct markers. The Hox A3 and Hox A4 are mainly longitudinal divergent genes in the endostyle. A total of 104 OSGs were defined as organ markers of the endostyle in *S. clava*. The expression similarity between the endostyle and the organs of *D. rerio,* including the gill and the head kidney, was proved in the OSGs-based cross-species alignment and organ transcriptomic profile comparison. The similarity between the endostyle and human thyroid/the stomach was proved to be of the highest rank in human tissue-enriched marker-based similarity calculations. Adapting to the structure consisting mainly of epithelial cells, similarities to the fallopian structure and skin were also seen. Interestingly, high similarities were also obtained in the comparison of endostyle against parathyroid, choroid, bone marrow, and retina. These similarities of the endostyle against organs from *D. rerio* and *H. sapiens* imply the potential function homolog of the pharynx organs and hematopoietic system of the endostyle.

## Figures and Tables

**Figure 1 biology-12-00245-f001:**
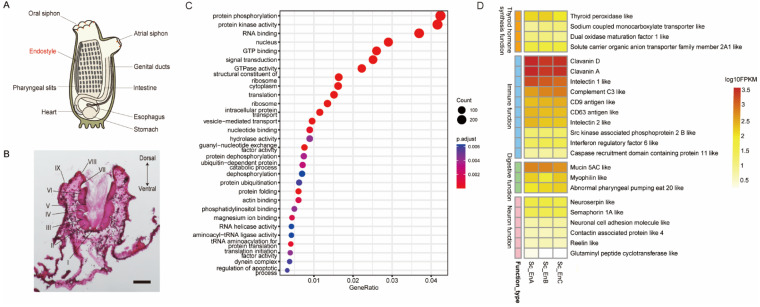
Morphology and transcriptional profile of ascidian endostyle. (**A**) Location of endostyle in adult ascidian; (**B**) HE staining on cross-sections of endostyle in *S. clava*. The relative positions of nine regions were labeled. Bar: 100 μm; (**C**) top expressed gene terms of endostyle by GO enrichment analysis. (**D**) Heatmap of top functional gene expression level for four main functional categories of the endostyle.

**Figure 2 biology-12-00245-f002:**
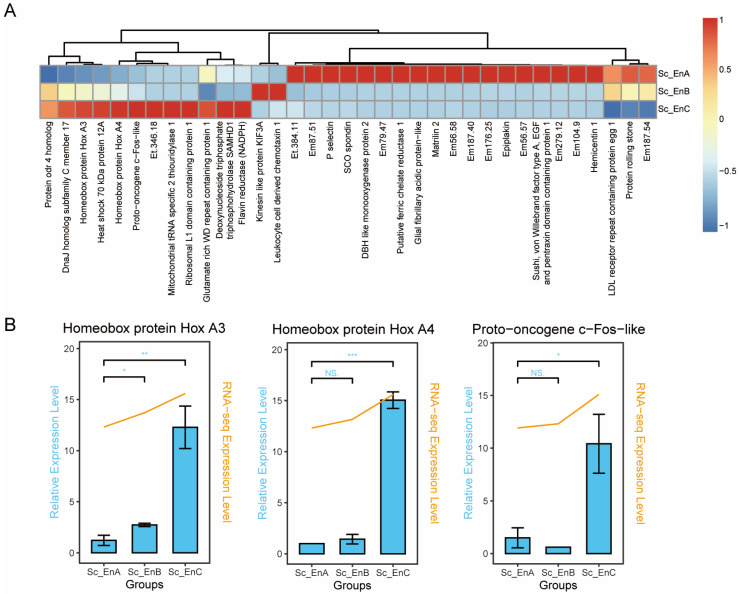
Identification of DEGs on endostyle longitudinal levels. (**A**) The expression level of up- or down-regulated genes was visualized with a heatmap for longitudinal segmented samples. Genes without functional annotation were labeled as shorted gene IDs with “Em” or “Et” starting. (**B**) qRT-PCR verifies the expression trend of DEGs in longitudinal levels of the endostyle. The blue bar corresponds to the relative expression level in qRT-PCR verification, based on which the significant signal was labeled. The orange broken line corresponds to the RNA-seq expression level in transcriptomic data. * *p* < 0.05. ** *p* < 0.01.*** *p* < 0.001. NS Not significant.

**Figure 3 biology-12-00245-f003:**
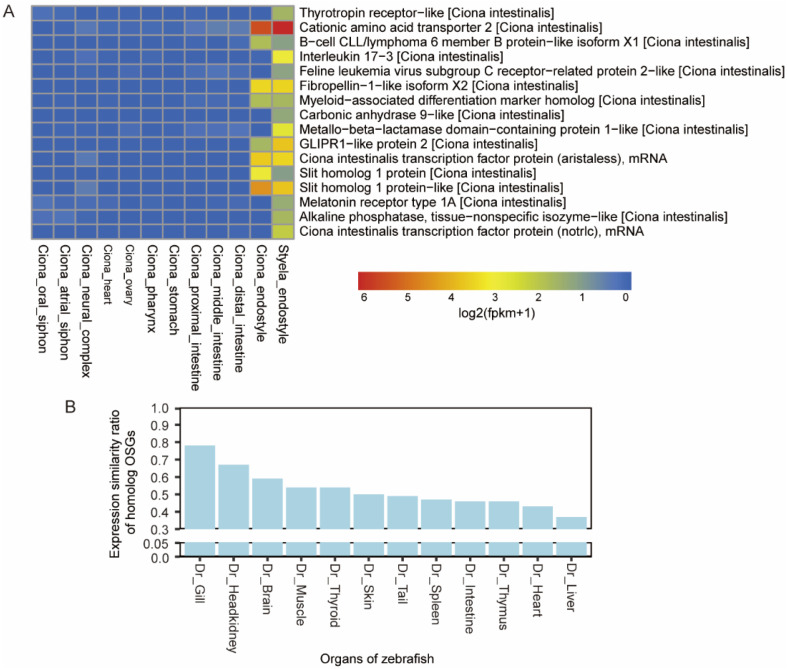
OSG-based expression similarity analysis. (**A**) Identification of OSGs in the endostyle of *S. clava*, based on the comparison with *C. robusta* organ or tissue RNA-seq results (except *Ciona* endostyle). Homolog gene pairs of *Ciona* and *Styela* with expression values less than 0.5 in the *Ciona* organs while expression values larger than 1 in *Styela* endostyle were defined as OSGs in the endostyle of *S. clava*; (**B**) expression similarity ratio calculation based on expression ratio of endostyle OSG homolog in *D. rerio* organs. The ratio was defined as the percentage of expressed endostyle OSG homolog (TPM larger than 1) in the expression profile of a *Danio* organ. For instance, for the gill in *Danio*, more than 70 percent of OSG homologs in *Danio* were expressed in the gill.

**Figure 4 biology-12-00245-f004:**
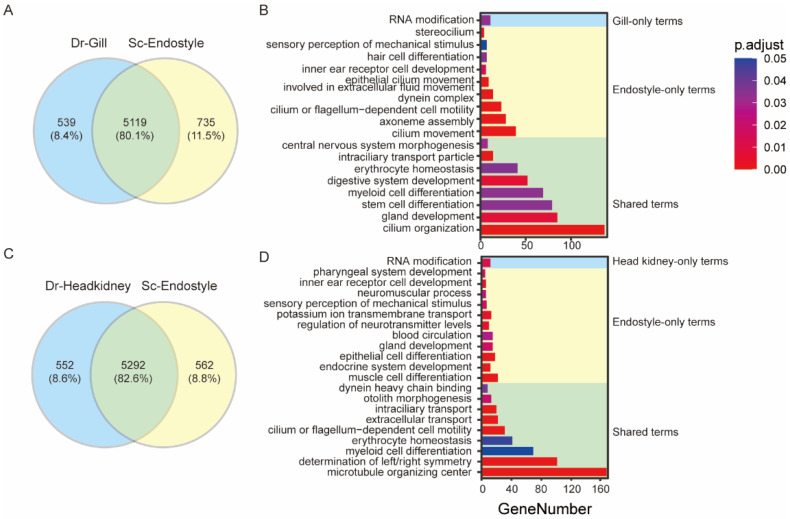
Expression profile comparison between endostyle and potentially similar organs. (**A**,**C**) Venn diagrams are patronizing the shared and unique expressed genes between the endostyle and the *Danio* gill and the *Danio* head kidney, respectively. (**B**,**D**) GO enrichment analysis for shared or organ specifically expressed gene lists in (**A**,**C**).

**Figure 5 biology-12-00245-f005:**
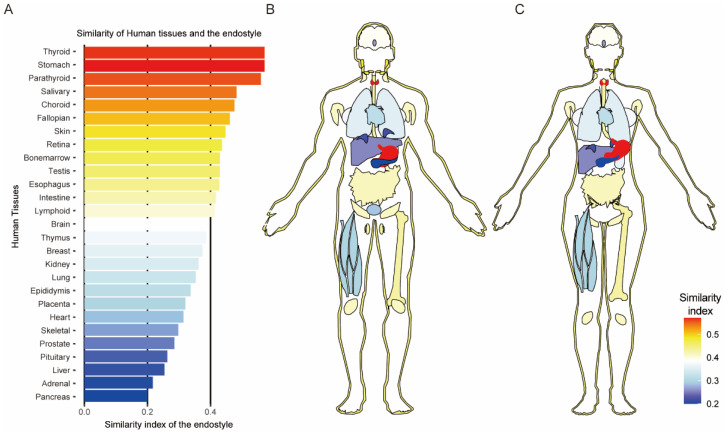
Transcriptional profile similarity between human tissue and the endostyle in *S. clava*. (**A**) Bar plot for similarity index in human organs corresponding to the endostyle; the similarity index is defined as the ratio of the highly expressed gene (FPKM > 5) of human organ-enriched marker gene homolog in the endostyle; if all homolog genes of human organ-enriched markers expressed in the endostyle with FPKM are larger than 5, the similarity index should be 1; (**B**,**C**) are similarity index projection on human organ anatomical graph.

**Table 1 biology-12-00245-t001:** Primer sequences used in the RT-PCR.

Gene Name	Forward Primer (5′-3′)	Reverse Primer (5′-3′)	Purpose
Homeobox protein Hox A3	ATGTGTAGCTATCAATATATGTATGATG	CTTTGCTAATACCGTCAGACCC	RT-PCR
Homeobox protein Hox A4	ATGGCTGCGTTACATGATG	AGATTCGGGAGAGCCCC	RT-PCR
Proto-oncogene c-Fos-like	GCTGCCAAAGGATTGCCTTC	TCTGTCCCTTCGTCTTTGCC	RT-PCR
Sc-18s	CTGAGTGAAGCAGCGAGTGTCTAACCTA	GCAAGTCCCTATCCCAATCACGAA	RT-PCR

## Data Availability

The genome sequences of *S. clava* were deposited in NCBI (BioProject number PRJNA523448). The transcriptome data of *S. clava* used for expression analysis were also deposited in the NCBI (BioProject number PRJNA913677).

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
