# Peer review of "Transcriptional Analysis of the Endostyle Reveals Pharyngeal Organ Functions in Ascidian"

_biology, 2023, doi:10.3390/biology12020245_

Round 1

Reviewer 1 Report

The authors used bulk RNA sequencing to explore the expression profiles of the endostyle of an ascidian, and implied that it shared functional similarities with the parathyroid, the hematopoietic system and the neuron system. This research presented interesting findings of the endostyle functions, and provided insights into the evolutionary origins of the vertebrate hormone secretion organs.

I think the following comments may help the authors to improve the manuscripts, which may give the audience a better understanding of what the authors did and the implications of their work.

Major comments:

1) The gill might have some functional similarities with the endostyle. How is the differences and similarities between the gill and the endostyle of Styela clava on transcriptomic levels?

2) The authors compared the endostyle with various human tissues and organs, but it appeared that the human thymus was not included? There could be similarity between the endostyle and the vertebrate thymus.

3) Bony fish like zebrafish are the only lineage having both the gill and the thymus? And in the GenBank, there may be some gill and thymus transcriptomic data for zebrafish. It would be interesting if a comparison between zebrafish gill/thymus and the ascidian endostyle.

Minor comment:

1) Line 478: Authors mentioned that Hox A3 had the highest expression at the posterior part of the endostyle”, and it seems the Hox3 was the lowest expression in the anterior (Fig. 2B). So how do explain “Hox A3 has stronger expression in the anterior....?  

2) Line 392:Authors have identified endostyle as homologous to some organs of other animals, and slit seems conserved in C. robusta and D. rario. Did the authors find this gene in humans?

3) Line 430: According to the morphological observation, the endostyle of ascidian was adjacent to the gill slit. Did the authors compare the different or common features of the filtering function between them?

4) Line 253-288, 297-397, 317-324, 414-426: No figures were cited in these sections (e.g. Fig 1C?), please check the full manuscript again and completed them.

5) Line 330, 365, 397: Please provide the IDs of the dataset the authors used.

Author Response

see attached document

Reviewer 2 Report

Major comments:

1)     Matsubara et al. (2021, Plos One) have done a similar but more comprehensive analysis on 11 different tissues of Ciona robusta, the authors should clarify why their study is necessary and what new findings they have made then.

2)     Without data from other tissues to compare, results and conclusions in section 3.1 seemed quite objective and inconvincible.

3)     In section 3.3, the authors identified endostyle OSG genes as those which are expressed (FPKM) less than 0.5 in all non-endostyle organs of C. robusta but higher than 1 in the endostyle of S. clava. It might be more reliable to do this using data all from C. robusta, to avoid batch effect.

4)     In section 3.4, the authors found that zebrafish brain and head kidney showed the highest expression similarity ratio to Ciona endostyle. And according to this, they indicate that the endostyle shares strong similarity to neuron and hematopoietic systems. This might be problematic, since only transcriptome data from several zebrafish non-thyroid organs were compared in their analysis. Moreover, the result and conclusion drawing here look very different from those in the following section when data from human and Ciona tissues were compared. I therefore suggest to remove this section.   

5)     In section 3.4 and 3.5, I suggest to determine Ciona-zebrafish and Ciona-human gene pairs first, and then to see how each zebrafish and human genes which are homologous to Ciona endostyle OSG genes are expressed. Moreover, a more comprehensive method like orthofinder, but not BLASTP, should be used to determine the relationship between Ciona genes and human/zebrafish genes.

Minor comments:

P12: I suggest to add “the basal vertebrate” before cyclostomate

P80-81: the description “other existing cyclostomes besides lamprey and hagfish” is incorrect, since no other living animals except for lamprey and hagfish belong to cyclostomes and the reference [17] the authors cited is actually talking about the direct development of thyroid gland in hagfish.

P249: nice >> nine

Fig1B should be marked in more details to show the nine regions of the endostyle. 

P430: “proto-vertebrate” should be protochordates and basal vertebrates

Author Response

Reviewer 3:

  1. When describing the similarity of human tissues and the endostyle, highly

expressed human enriched homolog in the endostyle should also be listed as a

separate file/resource not only the similarity index. The exhibition of these genes

will better support the functional similarity of human tissues and the endostyle.

Response: Thanks for your suggestion. The revised supplementary table of highly expressed human enrich homolog in the endostyle was listed in revised supplementary table S3. Thanks for your comments in enriching the supporting information of the manuscript.

  1. In Figure 5, how did you defined the threshold (FPKM > 5) for filtering a highly

expressed gene? In the analysis of transcriptional data, commonly believed

threshold is FPKM larger than 1.

Response: It’s a good question as the selection of the threshold is manually decided without a detailed explanation. For the highly expressed tissue-enriched genes selection in the endostyle, we tested different thresholds including FPKM=1, 5 or 10. The threshold set as 5 can best exhibit similarity differences in various human organs, while other thresholds may produce many similar similarity indexes.

  1. The title of the table 1 show be “Primer sequences used in the RT-PCR”. And some

of the primer sequences in the table 1 were not used in the afterwards experimental

results. The proper citation of figure 1C, 1D, and 2B were missed in the main result

paragraphs of the manuscript

Response: Thanks for your suggestions. False in the primer sequence table was corrected. The missed citation on figures was completed for all figures of the revised manuscript.

Round 2

Reviewer 2 Report

The authors have well addressed my comments and I have no further questions.